# Soy-Based Infant Formula is Associated with an Increased Prevalence of Comorbidities in Fragile X Syndrome

**DOI:** 10.3390/nu12103136

**Published:** 2020-10-14

**Authors:** Cara J. Westmark, Chad Kniss, Emmanuel Sampene, Angel Wang, Amie Milunovich, Kelly Elver, David Hessl, Amy Talboy, Jonathon Picker, Barbara Haas-Givler, Amy Esler, Andrea L. Gropman, Ryan Uy, Craig Erickson, Milen Velinov, Nicole Tartaglia, Elizabeth M. Berry-Kravis

**Affiliations:** 1Department of Neurology, University of Wisconsin, Madison, WI 53706, USA; 2Survey Center, University of Wisconsin, Madison, WI 53706, USA; ckniss@ssc.wisc.edu (C.K.); kelver@ssc.wisc.edu (K.E.); 3Department of Biostatistics & Medical Informatics, University of Wisconsin, Madison, WI 53792, USA; sampene@biostat.wisc.edu; 4Department of Pediatrics, Rush University Medical Center, Chicago, IL 60612, USA; angel_wang@rush.edu (A.W.); elizabeth_berry-kravis@rush.edu (E.M.B.-K.); 5National Fragile X Foundation, Folsom, CA 94596, USA; amie@fragileX.org; 6MIND Institute and Department of Psychiatry and Behavioral Sciences, University of California, Davis, CA 95817, USA; drhessl@ucdavis.edu; 7Departments of Human Genetics and Pediatrics, Emory University, Atlanta, GA 30322, USA; amy.talboy@emory.edu; 8Boston Children’s Hospital, Boston, MA 02115, USA; jonathon.picker@childrens.harvard.edu; 9Autism & Developmental Medicine Institute, Geisinger Lewisburg, Lewisburg, PA 17837, USA; bahaasgivler@geisinger.edu; 10Department of Pediatrics, University of Minnesota, Minneapolis, MN 55454, USA; else0007@umn.edu; 11Children’s National Health System, Washington, DC 20010, USA; AGropman@childrensnational.org (A.L.G.); RSUY@childrensnational.org (R.U.); 12Cincinnati Children’s Hospital Medical Center, Cincinnati, OH 45229, USA; craig.erickson@cchmc.org; 13Institute for Basic Research in Developmental Disabilities, Staten Island, NY 10314, USA; milen.velinov@opwdd.ny.gov; 14Department of Pediatrics, University of Colorado, Aurora, CO 80045, USA; Nicole.tartaglia@childrenscolorado.org

**Keywords:** autism, fragile X syndrome (FXS), infant formula, seizures, soy

## Abstract

A large number of adults and children consume soy in various forms, but little information is available regarding potential neurological side effects. Prior work indicates an association between the consumption of soy-based diets and seizure prevalence in mouse models of neurological disease and in children with autism. Herein, we sought to evaluate potential associations between the consumption of soy-based formula during infancy and disease comorbidities in persons with fragile X syndrome (FXS), while controlling for potentially confounding issues, through a retrospective case-control survey study of participants with FXS enrolled in the Fragile X Online Registry with Accessible Research Database (FORWARD). There was a 25% usage rate of soy-based infant formula in the study population. We found significant associations between the consumption of soy-based infant formula and the comorbidity of autism, gastrointestinal problems (GI) and allergies. Specifically, there was a 1.5-fold higher prevalence of autism, 1.9-fold GI problems and 1.7-fold allergies in participants reporting the use of soy-based infant formula. The major reason for starting soy-based infant formula was GI problems. The average age of seizure and allergy onset occurred long after the use of soy-based infant formula. We conclude that early-life feeding with soy-based infant formula is associated with the development of several disease comorbidities in FXS.

## 1. Introduction

Fragile X syndrome (FXS) is the most common form of inherited intellectual disability with a frequency of 1 in 5000 males and 1 in 4000–8000 females [1]. The disorder is clinically characterized by highly variable cognitive disability, autism, seizures, delays in language development, anxiety disorders, aggression and attention-deficit/hyperactivity disorder (ADHD) [1,2]. Seizures are the most substantial medical problem in children with FXS occurring in ~8–16% of males and ~3–7% of females, typically in the first 5 years of life [1]. Although FXS is not one of the 29 core conditions included in the newborn screening (NBS) guidelines developed by the American College of Medical Genetics, it is a high priority genetic disorder for which screening would be possible if there was an empirically-supported therapy [3]. Findings from our laboratory indicate that soy-based diets increase seizure prevalence in a mouse model of FXS (*Fmr1^KO^* mice) and are associated with increased febrile seizures, simple partial seizures, epilepsy comorbidity and autism phenotypes in a population of children with autism [4,5,6]. Thus, we hypothesized that consumption of soy-based formula during infancy exacerbates seizures in neurodevelopmental disorders such as FXS.

There is a dearth of studies regarding the effects of soy consumption on infant development [7,8,9,10,11,12,13]. Soy contains high levels of plant estrogens (phytoestrogens, isoflavones), which may mimic or antagonize natural estrogen activity. A substantial percentage (12%) of infant formulas are soy-based and have phytoestrogen levels in the range of 4.5–8 mg/kg/day [7,8,9,10,11]. Taking into consideration body weight, infants fed soy-based formulas consume 6–11 times the amount of phytoestrogens necessary to produce hormone-like effects in adults [14].

Current public health policies regarding soy-based infant formulas include positions from the American Academy of Pediatrics, “There is no conclusive evidence from animal, adult human, or infant populations that dietary soy isoflavones may adversely affect human development, reproduction, or endocrine function,” and the National Toxicology Program (NTP) Center for the Evaluation of Risks to Human Reproduction (CERHR), “The overall evidence was considered insufficient to reach a conclusion on whether the use of soy infant formula produces or does not produce developmental toxicity with infant exposure in girls or boys at recommended intake levels” [15,16]. These policies are in place for the general population, albeit there have been no studies specifically testing the effects of soy-based infant formulas in neurodevelopmental disabilities. Vulnerable populations, such as FXS, are likely more susceptible to the potential neurotoxic effects of high doses of bioactive dietary components.

It is not possible, at this time, to conduct a prospective evaluation of infant diet on FXS phenotypes as the average age of diagnosis is 35–37 months in boys and 42 months in girls [17], which occurs long after children have transitioned from formula to solid food. If specific food products are determined to affect the prevalence or severity of seizures or other disease phenotypes in FXS, NBS could be employed to identify susceptible infants and inform decisions regarding infant feeding recommendations. We conducted a retrospective survey analysis to determine if there were associations between the consumption of soy-based infant formula and seizure history, cognitive ability and autistic behaviors in participants with FXS enrolled in the Fragile X Online Registry with Accessible Research Database (FORWARD), the largest registry of FXS participants [18]. Previous FORWARD studies had not examined the impact of infant diet on disease outcomes. This analysis specifically examines associations between caregiver-reported use of soy-based infant formula and comorbid disorders, while comparing findings to prior data attained from the Simons Foundation Autism Research Initiative (SFARI) medical record database. We find significantly increased comorbidity of autism, GI problems and allergies in the FORWARD population associated with the use of soy-based infant formula. We emphasize that this study shows associations between soy-based infant formula and FXS comorbidities and not cause and effect relationships.

## 2. Materials and Methods

Ethics Approvals. Each participating clinic obtained institutional review board (IRB) approval from their institution before enrolling families in FORWARD. Those approvals provided that the clinics were allowed to contact families for participation in other research studies based on FORWARD consent form. Thus, this study did not need to be evaluated for IRB approval at the participating clinics as each clinic only provided packets to the families, who then contacted the University of Wisconsin Survey Center (UWSC) to participate. This study was reviewed by the UW Health Sciences IRB and determined to meet the criteria for exempt Human Subjects research in accordance with Category 2 defined under 45 CFR-46.

Study Design. We utilized a national registry of FXS families maintained by FORWARD to conduct the first case-control study evaluating associations between early childhood feeding practices and the severity of common FXS phenotypes (seizures, cognition and autistic behavior). The specific components of the study included the following: (1) design a questionnaire to assess demographics, infant feeding practices, frequency and severity of seizures, cognitive ability, autistic behaviors and comorbid diagnoses in a FXS population by parental survey; (2) recruit seizure (cases) and non-seizure (controls) full-mutation FXS participants to a retrospective case-control study examining associations between soy-based infant formula use and FXS phenotypes; and (3) examine associations between soy-based infant formula and clinical characteristics (i.e., frequency and severity of seizures, cognitive ability, autistic behaviors) while accounting for duration of use of soy-based infant formula and potentially confounding factors, such as food allergies. Hypotheses included the following: (1) soy-based infant formula will be associated with increased frequency and severity of seizures in FXS, (2) soy-based infant formula will be associated with decreased cognitive ability in FXS, and (3) soy-based infant formula will be associated with elevated autistic behaviors in FXS.

Questionnaire. A 55-point questionnaire was designed to assess demographics, infant feeding practices, frequency and severity of seizures, and comorbid diagnoses. Questions were adapted from the Center for Disease Control and Prevention (CDC) Infant Feeding Practices II, the Fragile X Clinical & Research Consortium (FXCRC) parent report form, and the SFARI medical records database questionnaires. In addition, cognitive ability and autistic behaviors were assessed by adapting questions from well-established tools into the questionnaire. Cognitive tools included the Ages and Stages Questionnaire (ASQ) for 36-months, and autistic tools included the Modified Checklist for Autism in Toddlers, Revised (M-CHAT-R), and the Social Responsiveness Scale (SRS) parental survey. The primary endpoint of interest examined was seizure history. The primary predictor variable was the type of infant milk (casein, soy, breast) consumed. Questions regarding variables of interest were embedded in the survey to discourage participant knowledge of the study hypothesis. Demographic information on gender, age, race/ethnicity was collected. Existing data in FORWARD could not be used in this study because their coordinating center is not allowed to provide data at an individual level to investigators due to the rarity of the disorder.

Study Population & Participant Recruitment. The FXCRC was established in 2006 with support from the National Fragile X Foundation (NFXF) and subsequently expanded in 2009 with support by a grant from the Centers for Disease Control and Prevention (CDC) and consists of 22 FXS clinics and research facilities across the United States. The FXCRC developed FORWARD in 2011 to facilitate multisite data collection on individuals with FXS and to assist researchers in identifying participants who may be interested in and meet the eligibility criteria for specific research projects [18]. More than 1350 of the registrants are full-mutation FXS, and 1102 have a completed Clinical Report Form with seizure history. The study population included full-mutation FXS individuals enrolled in FORWARD and whose parents/caregivers had previously agreed to be contacted for research studies. Per FORWARD internal review board (IRB) guidelines, participants must be contacted directly by clinic directors for participation in research studies. Thus, for recruitment, participants in FORWARD with a history of seizures were identified as well as 4-fold more control participants. Controls were FXS participants without a reported history of seizures. Cases and controls were reasonably balanced on age and sex. The Principal Investigator, at the UW-Madison, contacted the clinical directors at all FORWARD sites and invited them to participate in the study. Individual clinics that agreed to participate in the study were provided a list of eligible participants associated with their site by the staff at FORWARD. The clinics sent parents/guardians a letter that informed them of the proposed study, invited them to participate, and emphasized the voluntary and confidential nature of the research. The letter requested that the primary caregiver of the participant with FXS return an enclosed card with their contact information to the UWSC for participation in the project.

Recruitment Sample Size Justification. During the study design phase, FORWARD had a total of 122 enrolled participants with a seizure history (cases). By recruiting all cases with a seizure history (*n* = 122) and 4-fold more non-seizure controls (*n* = 488) to the study, estimating a 50% response rate (*n* = 61 cases; *n* = 244 controls), and assuming that the non-seizure population had a 20% usage rate of soy-based infant formula similar to the SFARI population [5], the proposed sample size would be able to detect with 80% (90%) power a 16 (19) percentage point difference in the proportion of soy-based formula use among cases versus controls, with a one-sided test of type I error of 0.05. Based on prior recruiting studies through FORWARD, a 50% response rate was deemed very conservative. Parents of FXS individuals are a highly motivated group that enthusiastically volunteer to participate in research projects to further therapeutic interventions for their children. There is a long history of collaboration and trust between these families and the clinician members of FORWARD that sent out the initial invitations for study participation. An expectation of 36% (39%) soy use among cases was reasonable as previous analysis of autism participants in the SFARI database indicated a 44% rate of soy-based infant formula use in participants reporting simple partial seizures [5].

Data Collection. Data collection occurred in two phases because the identities of FORWARD participants were not directly available to the Principal Investigator and included a four-wave mail survey strategy to ensure maximum response rates. Phase 1 consisted of sending prepared sample collection packets to 10 FXS clinics across the United States. The packets were designed to only need a printed address label attached to each packet (postage was already provided for each packet) A copy of the packet that was used for Rush University Medical Center is provided in Appendix B. Phase 2 consisted of (1) mailing the questionnaire (*Fragile X Syndrome Nutrition Study*, Appendix C) with a cover letter from the Principal Investigator containing a $2 bill pre-incentive and a postage-paid return envelope; (2) mailing a thank you postcard reminder 5–7 days after the initial mailing; and (3) sending full mailings (same as the first, but without the $2 bill and a slightly differently worded cover letter) to non-responders from the first two mailings 3–4 weeks after the initial survey packets were sent. The UWSC mailed the questionnaires and incentives, tracked responses, and provided an electronic dataset to the Principal Investigator. The informed consent document was embedded in the questionnaire to ensure return of the signed documents.

Data Analysis. Data were analyzed in accordance with STROBE guidelines. Percentages, means, standard error of the means (SEM), odds ratios (OR), and 95% confidence intervals (CI) were computed to describe the population. Fisher exact test (if less than 5 outcomes per cell) or Pearson’s uncorrected chi-square tests were used to examine the null hypotheses that the prevalence of comorbidities in FXS are the same in infants fed soy-based infant formula or not. Student’s *t*-tests were used to compare the means of two populations. Statistical significance was defined as *p* < 0.05. A Bonferroni correction was not applied for multiple comparisons. The number of participants for each comparison is reported in the corresponding tables. This manuscript addresses the hypothesis that soy-based infant formula is associated with increased frequency and severity of seizures in FXS. Subsequent manuscripts will address the other study hypotheses regarding cognitive ability and specific autistic behaviors as well the association of comorbidities with breast milk.

## 3. Results

### 3.1. Response Rates

Between 27 August 2019 and 6 November 2019, the UWSC mailed out a total of 863 sample collection packets to the 10 participating FXS Clinics (Table 1). A total of 185 caregivers returned participation cards requesting a total of 241 questionnaires for persons with FXS (some families had multiple children with FXS). Thus, the overall phase 1 response rate for the study was 21%. Response rates for individual clinics varied between 0–54% (Table 1).

Phase 2 of the data collection involved sending the 12-page questionnaire (*Fragile X Syndrome Nutrition Study*) to recruited participants. Between 19 December 2019 and 7 February 2020, the UWSC mailed out a total of 234 survey packets to the 180 respondents from phase 1. A copy of the questionnaire is provided in Appendix C. During phase 2, the UWSC learned that 11 of the 234 mailed survey packets were duplicates of other survey packets. At the end of the data collection period (28 February 2020), the UWSC estimated that they sent survey packets to a total of 223 non-duplicated parents or guardians resulting in 199 returned non-duplicated questionnaires. There were 9 pairs of participants with identical birth dates, but each pair differed on a defining characteristic such as sex, autism comorbidity, allergy comorbidity, or birth weight indicating a low likelihood of participant duplication in the dataset. The overall phase 2 response rate for the study was 89% (199 returned surveys per 223 non-duplicated cases) (Table 2). Seven of the 10 clinics had a 75% or greater response rate for phase 2, which is outstanding for a mail survey.

### 3.2. Participant Demographics

The population for this study included full-mutation FXS registrants in FORWARD. Participants were invited based on a case-control study design, which included all registrants with a seizure history (cases) and 4-fold more participants without a seizure history (controls). At the end of the field period, there were 199 completed surveys. Based on caregiver-reported seizure history, the final dataset included 28 cases and 169 controls equating to 14% of participants reporting a seizure history. Demographics indicated 73% male participants, which coincides with the reported enrollment of 76% full mutation males in FORWARD [18]. The cases and controls had similar characteristics regarding ethnicity, birth weight and length, and current BMI (Table 3). Participant age at the time of the survey was significantly higher, 23 versus 17 years, for the cases; however, the average reported age of the first seizure was 7 years (SEM 0.97). Exclusion of participants under 7 years of age from the control cohort resulted in an average age of 19 years (SEM 0.78), which is not statistically different from the cases (*p* = 0.09). The prevalence of comorbidities, including autism, Down syndrome, food allergies, diabetes, and gastrointestinal (GI) problems, was not significantly different between cases and controls. Reported epilepsy comorbidity was inconsistent with reported and clinical seizure history and was not included in this analysis.

### 3.3. Infant Feeding

In the study population, 142 out of 199 caregivers indicated that their child had been fed breastmilk, 52 respondents checked “No” to Q18, 2 respondents left the question blank, and 3 respondents checked “Don’t know”. This corresponds to a 73% rate of breastfeeding among Yes/No respondents, which aligns with CDC data indicating national averages from 76–84% between 2009–2016 in the general population [19]. In the study population, 96 out of 199 caregivers indicated that their child had been fed cow milk formula during their first year of life, 90 respondents checked “No” to Q23, 1 respondent left the question blank, and 12 respondents checked “Don’t know”. This corresponds to a 52% usage rate of cow milk formula among Yes/No respondents. In the study population, 45 out of 199 caregivers indicated that their child with FXS had been fed soy-based formula during their first year of life, 132 respondents checked “No” to Q28, 3 respondents left the question blank, and 19 respondents checked “Don’t know”. This corresponds to a 25% usage rate of soy-based infant formula among Yes/No respondents compared to 17% in the SFARI population [5] and 12% in the general population [5,6,7,8,9,10,11,12,13,14,15,16]. Binning the infant feeding data by cases and controls indicated no statistically significant differences dependent on infant diet (Table 4). The pre-study power calculation was based on recruitment of 305 participants and assumed a 16–19-percentage point difference in the use of soy-based infant formula between cases and controls. The study recruited at 65% (199 participants) of the anticipated level. The non-significant 1.7-fold increase in the use of soy-based infant formula in the cases was based on a 16-percentage point difference (*p* = 0.09).

### 3.4. Comorbid Conditions as a Function of Infant Feeding with Soy-Based Formula

There was a 1.5-fold higher rate of comorbid autism in FXS participants who had been fed soy-based infant formula (64% autism with soy versus 44% no soy; *p* = 0.026) (Table 5). These findings are congruent with cited prevalence rates of 15–67% of comorbid autism in FXS [20,21]. The prevalence of food allergies was not statistically different between soy and no soy cohorts, albeit a larger study population may statistically support a 2-fold increase in food allergies associated with soy-based infant formula in FXS. The prevalence of diabetes was very low with only 2 participants out of 199 reporting diabetes. This was expected as diabetes is very rare in FXS [22]. There was a 1.9-fold higher rate of GI problems in FXS participants who had been fed soy-based infant formula (47% GI problems with soy versus 25% no soy; *p* < 0.01). The vast majority of respondents reported GI problems commencing before 3 years of age. GI problems are common in children and adults with FXS [23,24]. There was a 2.0-fold (not statistically significant) higher rate of reported seizures in the FXS participants who had been fed soy-based infant formula (*p* = 0.08). These data support findings from the SFARI autism population where there was a 2.1-fold higher rate of comorbid epilepsy in a high functioning autism population who had been fed soy-based infant formula (3.6% epilepsy with soy, *N* = 330, versus 1.7% no soy, *N* = 1563; *p* = 0.02) [5]. There was a 1.7-fold higher rate of reported allergies in FXS participants who had been fed soy-based infant formula (56% allergies with soy versus 33% no soy; *p* < 0.01). These data corroborate findings from the SFARI autism population where there was a 2.3-fold higher rate of allergies with soy-based infant formula (3.5% allergies with soy, *N* = 341, versus 1.5% no soy, *N* = 1608; *p* = 0.01), albeit the overall allergy prevalence was much higher in the FXS population [25]. Participants (8.3%) in the soy cohort reported no comorbid conditions compared to 33% in the no soy cohort (*p* = 0.004). Data binned as a function of sex indicate similar trends for males and females (Appendix A). Overall, these data demonstrate that soy-based infant formula is associated with several disease comorbidities in FXS.

### 3.5. Seizure Types and Timing in the Study Population

The primary hypothesis to be tested in this study was that the prevalence of seizures in FXS would be higher in infants fed soy-based versus non-soy-based formula or breast milk. Prior to commencing the study, FORWARD staff estimated that approximately 11% of registrants had a seizure history. The study design included recruitment at one case per 4 controls to increase participation by participants with a seizure history. Based on caregiver-reported seizure history, the dataset included 28 cases and 169 controls, or 14% of participants reporting a seizure history. Thus, the study did not succeed in enriching for participants with a seizure history. Regarding seizure type, 9 participants reported data regarding both seizure type and the use of soy-based infant formula and 16 participants for seizure type and no soy use (Table 6). Of note, no participants reported febrile seizures, which were reported in 2% of the SFARI population where they were 2.6-fold higher in participants fed soy-based infant formula (soy 4.2%, *N* = 333; no soy 1.6%, *N* = 1581) [5]. Based on concerns that seizures types reported by caregivers did not reflect prior literature on seizure types in FXS [1], 17 patients from Rush with seizures could be identified and compared with data regarding seizure type in the medical record. Only 4 families correctly identified the seizure type and thus data from the survey regarding seizure type cannot be considered accurate.

Data regarding the age at which participants with FXS had their first seizure was available for 26 participants (mea*N* = 7.5 years; SEM = 0.97) and was not statistically different between soy-based infant formula (mea*N* = 8.2 years; SEM = 2.2; *N* = 9) and no soy based-infant formula (mea*N* = 7.0 years; SEM = 1.0; *N* = 15; *p* = 0.58). Twenty-six caregivers answered the question, “Was your child given medication to treat the seizures?”, with 22 reporting “Yes”. The participants not receiving medications were fed breast milk and/or cow milk formula (3 reporting). The participants medicated for seizures were fed breast milk (68%, *N* = 22 reporting), cow milk formula (45%, *N* = 20 reporting) and soy-based formula (43%; *N* = 21 reporting). Although not statistically significant with the small sample size, these data suggest that that use of soy-based infant formula may be associated with increased use of medications to treat seizures in FXS.

### 3.6. Caregiver-Reported Reasons for Starting and Stopping Soy-Based Infant Formula

In response to Q30, “Some examples of why people might choose to feed their child soy-based formulas include problems with other foods such as allergy, intolerance, constipation, diarrhea, too much mucus, gas, too much spit up, vomiting, or parental choice. Why was your child fed soy-based formula?”, the most common reasons included gastrointestinal issues (Table 7). The most common reasons for stopping soy-based infant formula were related to the child’s age and the transition from formula to whole cow milk and solid foods (Table 8).

### 3.7. Gastrointestinal Problems in the Study Population

Respondents (*N* = 57) provided data regarding the start of GI problems in their children with FXS. For 8 participants, data was not available regarding the use of soy-based infant formula. The remaining respondents reported use of soy-based infant formula (*N* = 18) or not (*N* = 31). All FXS participants reporting the use of soy-based infant formula and GI problems, with age data available for when GI problems started, commenced GI problems between birth and 3 years of age with 61% having GI problems within 2 weeks of age and 78% within the first year compared to only 9.7% within 2 weeks and 58% within the first year for participants not fed soy-based infant formula (Table 9). The average age of commencing GI problems was 7.8 months for the soy cohort compared to over 4 years for the no soy cohort. For all participants reporting GI problems within 2 weeks of age and the use of soy-based infant formula (*N* = 11), GI problems were reported before or at the same time as starting soy-based infant formula. These data suggest that soy-based infant formula is used at higher rates in FXS due to comorbid GI problems in newborns.

### 3.8. Allergies in the Study Population

Respondents (*N* = 73) reported data regarding the age of commencement and/or identification of allergies in their children with FXS. Allergies trended an average of 22 months earlier (*p* = 0.06) with a significant increase in prevalence within both 2 weeks and 1 year of age (*p* < 0.05) in FXS participants fed soy-based infant formula versus no soy (Table 10). In response to Q17, “Some examples of allergens include pollen, dust, pets, latex, eggs, fish, gluten or wheat, milk, nuts, medications, and many others. What are the allergens your child reacts to now or has in the past?”, the most prevalent allergens were seasonal related followed by food products and medications (Table 11). All participants reporting the timing of allergies exhibited symptoms by 13 years of age. Of note, all participants reporting allergies within 2 weeks of age included milk or lactose as an allergen. The percentage of participants with FXS reporting allergies (37%) is comparable to the general population in which 30% of adults and 40% of children have allergies. The percentage of participants with FXS reporting food allergies (16%) is higher than the general population, which reports 5.4–8% prevalence of food allergies in children [26,27]. Caregivers (*N* = 16, 8%) reported allergic reactions to Amoxicillin and/or cephalasporin antibiotics. These drugs are commonly prescribed for bacterial infections including otitis media (middle ear infections), which are common in FXS. About 10% of Americans report allergies to penicillin or related antibiotics [28]. Of 21 participants reporting use of soy-based infant formula, the prevalence of allergies and the timing of both, only 2 reported the prevalence of allergies before commencing soy-based infant formula and both of those reported milk allergies within 2 weeks of birth. The average age of allergy onset occurs long after discontinuation of infant formula.

## 4. Discussion

There is little knowledge regarding how early life feeding is associated with neurological development, particularly in neurodevelopmental disorders. To fill this gap, we conducted the first study examining associations between infant feeding practices, specifically soy-based infant formula, and disease outcomes in children with FXS using FORWARD as a sampling frame to collect new data through parental surveys. This unique study population, with a high prevalence of seizures compared to the general population, allowed for a smaller cohort size. Published estimates of formula intolerance range from 2–7.5%; yet, 12% of infants are fed soy-based formula suggesting that nonstandard, soy-based formulas are used excessively [16]. Infants with FXS are often hypotonic and have initial poor latch and suck with breastfeeding, as well as frequent recurrent emesis because of reflux [1]. We found that 25% of FORWARD participants in this survey study reported use of soy-based infant formula. We also found associations between the consumption of soy-based infant formula and increased comorbidity of autism, GI problems and allergies. The data did not reach statistical significance to corroborate prior retrospective analysis of medical record data from the SFARI population where the prevalence of seizures in autistic children fed soy-based infant formula was higher [5]. However, line item analysis of the Aberrant Behavior Checklist (ABC) in the SFARI population indicated that several autistic behaviors were exacerbated in autistic children reported to have consumed soy-based infant formula [6], and analysis of comorbid conditions in the SFARI population indicated associations between soy-based formula and increased allergies, ADHD and bipolar disorder [25]. Overall, these findings from FORWARD FXS and SFARI autism populations strongly suggest that early-life feeding with soy-based infant formula is associated with adverse neurological outcomes in these developmental disabilities. It remains to be determined if babies destined to have more severe disability (autism, seizures, etc.) are harder to breastfeed (will not latch on, hyper, colic) and have more feeding intolerance and potentially more severe problems with gastroesophageal reflux (GERD) and bowel dysregulation, leading to the increased use of formula including soy formula.

An important goal of this project was to account for potentially confounding issues instigating the use of soy-based infant formula. The Infant Feeding and Early Development (IFED) Study found that the top-rated maternal reasons for use of soy-based formula include: “I fed my other child(ren) soy formula” (54%), “I think soy is healthier than other types of formula” (54%), “I suspect my baby had milk intolerance or my family has trouble digesting cow’s milk” (52%), “My family and friends recommended it” (27%), “I chose soy formula for religious reasons” (17%), “Health care provider recommended soy” (15%), “I prefer a dairy-free diet” (12%), “My child had colic or other digestive problem” (7%), “I suspected my baby had a food allergy or intolerance (other than milk)” (7%), and “Family follows a vegan diet” (1%) [12]. In the IFED population, 70% of mothers were Black, and 57% had a high school education or less. In contrast, in FORWARD, the highest ranked reasons for the use of soy-based infant formula were medical-related, particularly GI issues. In the soy cohort, 61% of infants exhibited GI problems within 2 weeks of age, and the initiation of soy-based formula started at the same time or after the GI problems. These data suggest that a high rate of GI problems in newborns with FXS necessitates alternate feeding strategies including the use of soy-based infant formula. Thus, early GI problems are a potentially confounding issue here. It is possible that the GI issues and resulting effects on nutritional intake produced the increased prevalence of autism, and that soy is simply a marker (correlate) of the underlying cause of the GI problems.

The strengths of this study design include: (1) this study is the first to link parent-reported data on infant feeding practices of children with FXS to disease phenotypes in those children; (2) FORWARD offers a unique study population with a high prevalence of seizures compared to the general population allowing for smaller cohort sizes; (3) the caregivers are a highly motivated group of parents eager to participate in research studies; and (4) FORWARD is an established FXS registry.

The limitations of this study include the following: (1) A limitation of all retrospective studies is the inability to fully evaluate the temporality of the relationships examined. Although a longitudinal approach would better enable evaluation of causality, it is currently impossible in this population because most participants are not diagnosed with FXS until three years of age after infant formula has been discontinued. (2) Another limitation is recall bias. Parents were asked to answer a series of questions regarding infant feeding and the frequency and severity of seizures, which may have occurred years earlier. The selection of questions was chosen with great care as to yield more accurate measures, and recall bias regarding infant formula usage was not expected to be a problem as parents typically switch formulas for specific reasons such as GI problems or allergies. (3) Another limitation is parental report of seizure type, which was not accurate as families are not trained in classifying seizures and doctors may not have told the parents which type(s) of seizures their child had. These data are best attained by review of the medical records. (4) Another limitation is response rates leading to concerns that that the case sample is not representative of the population of children with FXS experiencing seizures. We observed a low response rate (19%) at the phase 1 level and a high response rate (89%) at the phase 2 level. The majority (61%) of the completed questionnaires were from a single clinic (Rush University Medical Center). We could not address the Phase 1 response rate issue due to FORWARD IRB regulations that precluded our inability to contact participants until after they returned contact information to the UWSC. Moving forward, we plan to increase participant response by conducting the survey online through the NFXF whereby consent is attained through voluntary participation in an online survey. (5) Another limitation is potential selection bias. Many participants switch from breast milk to cow milk formula to soy-based formula, and the need for multiple dietary changes may be due to a more severe FXS phenotype.

In a clinical setting, parents seek medical attention for their infants with FXS at the onset of developmental delays. Population-wide NBS is not performed in the U.S. because there is no treatment. Hence, most children are three years or older before diagnosis. Considering the dearth of treatments for FXS and the high comorbidity of early-life GI problems, our findings warrant further investigation. There are alternatives to soy-based infant formula that could be implemented in the clinic for infants with FXS experiencing problems with breastfeeding and cow milk-based formulas. However, NBS for FXS would be required to enable an early diagnosis and choice of infant feeding.

## 5. Conclusions

In conclusion, this case-control, retrospective survey study provides data regarding associations between the use of soy-based infant formula and comorbid conditions in participants with FXS recruited through FORWARD. Of importance, the data suggest that the usage rate of soy-based infant formula is twice as high in FXS compared to the general population and that the prevalence of autism, GI problems and allergies in FXS participants is statistically higher in soy-fed participants. It remains to be determined if soy-based infant formula causes increased comorbidities or serves as a correlate for underlying GI problems in FXS. The findings support further study examining the effects of infant nutrition on neurological development and consideration of the pros and cons of implementing NBS for FXS to identify individuals who may benefit from early nutritional management. Some mothers know that they are going to have a baby with FXS, and they could receive nutritional counseling now.

## Figures and Tables

**Table 1 nutrients-12-03136-t001:** Clinic Participation Phase 1.

Clinic	Sample Collection Packets			Response Rate (%) ^1^
	Sent	Returned	Duplicates	
Children’s Hospital Boston	36	3	0	8
Children’s Hospital Colorado	111	18	0	16
Children’s Nat Health System	13	3	0	23
Cincinnati Children’s	78	0	0	0
Emory Univ School of Medicine	44	11	0	25
Geisinger Medical Clinic	21	7	0	33
NY State Institute for Basic Research	50	13	1	26
Rush Univ Medical Center	375	106	14	28
UC Davis Health System	122	17	1	14
University of Minnesota	13	7	0	54
TOTAL ^2^	863	185	16	21

^1^ The response rate was calculated by dividing the number of returned sample collection cards by the number of sample collection packets sent out and multiplying by 100. ^2^ Five sample collection packets requesting questionnaires for 7 persons were returned too late (after 1 February 2020) to be included in the phase 2 mailing. The remaining 180 returned cards requesting questionnaires for 234 participants contained 16 duplicates whereby the same clinic sent multiple sample collection packets to the same parent/guardian (*N* = 14) or multiple clinics each sent a sample collection packet to the same parent/guardian (*N* = 2). To adhere to institutional review board (IRB) rules, the University of Wisconsin Survey Center (UWSC) did not see the list of 863 parents/guardians that received sample collection packets from the 10 fragile X syndrome (FXS) clinics. Thus, the total number of duplicates cannot be certain. Elimination of known duplicates and late returns gave a return rate of 19% (164/842*100).

**Table 2 nutrients-12-03136-t002:** Subject Participation Phase 2.

Clinic	Questionnaires			Response Rate (%) ^1^
	Requested	Sent	Returned	
Children’s Hospital Boston ^2^	3	0	0	0
Children’s Hospital Colorado ^2^	28	26	25	96
Children’s Nat Health System	4	4	4	100
Cincinnati Children’s	0	0	0	0
Emory Univ School of Medicine	10	10	9	90
Geisinger Medical Clinic	10	10	4	40
NY State Institute for Basic Research	15	15	13	87
Rush Univ Medical Center ^3^	142	142	130	92
UC Davis Health System ^2^	21	19	17	89
University of Minnesota	8	8	6	75
TOTAL ^2^	241	234	208	89

^1^ The response rate was calculated by dividing the number of returned questionnaires by the number of sent questionnaires and multiplying by 100. ^2^ There were 5 sample collection cards, requesting questionnaires for 7 persons with FXS, that were returned too late to be included in the phase 2 mailing. These participants were not sent questionnaires as the field period was designed to end by the end of February 2020, and there was not enough time to mail survey packets out and expect to have them returned. ^3^ Eleven of the mailed survey packets were duplicates of other survey packets that were not removed between phases 1 and 2. The response rate for Rush is the same (92%) after elimination of the duplicates resulting in 131 sent packets and 121 returned packets. The overall response rate was unchanged at 89%.

**Table 3 nutrients-12-03136-t003:** Characteristics of the FXS Population.

		Cases (Seizures)	Controls (No Seizures)
Population Size, % (*N*)	Male	16 (23)	84 (121)
	Female	9.4 (5)	91 (48)
Ethnicity, %	American Indian or Alaskan	0	0.59
	Asian	3.6	1.8
	Black or African American	0	4.1
	Hispanic or Latino	3.6	7.1
	Native Hawaiian ^1^	0	0
	White	100	91
	Other Race or Ethnicity ^2^	0	1.8
Age at Survey, Years (SEM) ^3^		23 (2.0)	17 (0.78)
Birth Weight, Pounds (SEM)		7.8 (0.25)	7.5 (0.10)
Birth Length, Inches (SEM)		21 (0.35)	20 (0.15)
Current BMI, Pounds (SEM)		24 (1.1)	23 (0.62)
Comorbidities, % (*N*)	Autism	64 (25)	47 (157)
	Down Syndrome	0 (28)	0 (158)
	Food Allergies	15 (26)	9.9 (152)
	Diabetes	3.6 (28)	0.64 (157)
	GI Problems	26 (27)	32 (164)

SEM: Standard error of the mean; GI: gastrointestinal; ^1^ Native Hawaiian or Other Pacific Islander. ^2^ Other Race or Ethnicity answers included 1 response “3/4 Eastern European Jew, ¼ Punjabi Indian”, 1 response “German and Scandinavian”, and 1 response “German and Norwegian”. ^3^ Student *t*-test, *p* = 0.007. The median age for cases was 20 and the median age for controls was 16.

**Table 4 nutrients-12-03136-t004:** Infant Feeding in the FXS Population.

	Cases ^3^ (Seizures)	Controls (No Seizures)	Chi Square *p*
Breast milk, % (*N*)	63 (27)	74 (165)	0.24
Cow milk formula, % (*N*) ^1^	50 (24)	50 (157)	0.98
Soy-based formula, % (*N*) ^2^	38 (24)	22 (148)	0.09

^1^ Participants (*n* = 1 case and *n* = 4 controls) were counted as “No” for cow milk formula because duration was 3 days or less. ^2^ Participants (*n* = 3 controls) were counted as “No” for soy-based infant formula because the duration was 3 days or less. ^3^ Participants (*n* = 1 case) was excluded because the seizure was due to sepsis at birth before feeding commenced.

**Table 5 nutrients-12-03136-t005:** Analysis of FXS comorbidities as a function of soy-based infant formula.

Phenotype	Soy % (*N*)	No Soy % (*N*)	*p* ^1^	Odds Ratio	95% CI
none	8.3 (36)	33 (105)	0.0040 ^2^	5.5	1.6–19
autism	64 (39)	44 (126)	0.026	2.3	1.1–4.8
food allergies	18 (40)	9.1 (121)	0.14	2.1	0.76–5.9
diabetes	0 (41)	1.6 (126)	1.0	0.60	0.028–13
GI problems	47 (43)	25 (129)	0.0073	2.6	1.3–5.4
seizures	22 (41)	11 (134)	0.08	2.2	0.90–5.6
allergies	56 (43)	33 (132)	0.0086	2.5	1.3–5.1

CI: Confidence interval; ^1^ Chi-squared test was used unless any variable contained less than *N* = 5 in which case Fisher exact test was used. ^2^ Fisher exact test.

**Table 6 nutrients-12-03136-t006:** Parent versus clinic-reported seizure types in study population.

Seizure Type	Parent-Reported		Clinic Chart ^5^	
	Soy % (*N*) ^1,2^	No Soy % (*N*) ^3,4^	Soy % (*N*) ^6^	No Soy % (*N*) ^7^
febrile	0 (6)	0 (12)	0 (3)	0 (6)
atonic	20 (5)	0 (11)	0 (3)	0 (6)
generalized	75 (4)	50 (12)	67 (3)	33 (6)
absence	60 (5)	70 (10)	0 (3)	0 (6)
simple partial	50 (4)	44 (9)	0 (3)	0 (6)
complex partial	50 (4)	20 (10)	67 (3)	67 (6)
infantile spasms	0 (5)	13 (8)	0 (3)	0 (6)

^1^ There was a total of 9 participants answering YES to use of soy formula and providing at least partial data for seizure type. The number of participants providing YES/NO data for each seizure type is provided in parentheses. Four participants reported multiple seizure types. ^2^ One participant reported “some other type of seizure”, which included “found in bed motionless and nonresponsive for 4–5 h twice”. ^3^ There was a total of 16 participants answering NO to use of soy formula and providing at least partial data for seizure type. The number of participants providing YES/NO data for each seizure type is provided in parentheses. Six participants reported multiple seizure types. ^4^ Three participants reported other seizure types, which were described as: “acting out”, “post birth due to sepsis (group B strep)”, and “one seizure followed by paralysis on one side”. The sepsis seizure was not included in the analysis as it occurred at birth before feeding commenced. ^5^ Clinical records from Rush University Medical Center were matched with 17 of 20 participants with a seizure history. Of the 17 chart reviews, parents correctly identified the seizure type in 4 surveys, identified a correct seizure type but either included additional seizure type(s) and/or missed a seizure type in 3 surveys, and wrongly identified the seizure type in 6 surveys. Caregiver-reported seizure data was missing for 4 participants. ^6^ There were 3 Rush participants answering YES, and 7 participants answering NO, to use of soy formula and having both parent- and clinic-reported seizure data. One participant reported both generalized tonic-clonic and complex seizures. ^7^ One participant reported another seizure type: “post birth due to sepsis (group B strep)”, and was not included in the analysis as the seizure occurred at birth before feeding commenced.

**Table 7 nutrients-12-03136-t007:** Reasons children with FXS were fed soy-based formulas.

*N*	Reasons ^1^
12	Vomiting, reflux, gas, diarrhea, constipation, colic
10	Spitting up
10	Allergy or intolerance to cow milk, rash
9	Doctor recommended or started in hospital
5	Parental choice; thought it was better, healthier, gentler on stomach
4	Not making enough breast milk, wouldn’t latch on
3	Available at the time, added food source
3	Irritable, fussy, ear infection
1	Liked soy more than others

^1^ A total of 45 participants provided responses with many participants listing multiple reasons.

**Table 8 nutrients-12-03136-t008:** Reasons children with FXS were stopped being fed soy-based formulas.

*N*	Reasons ^1^
21	Transitioned to whole cow milk and/or baby food, child age, drinking from cup
7	Didn’t help with vomiting, acid reflux, fussiness or colic or caused spitting up, vomiting, stomach pain or colic
5	Parental choice
4	Transitioned to soy milk
3	Doctor recommendation
2	Didn’t accept it
1	Outgrew reflux
1	Learned breast milk could be stored

^1^ A total of 41 caregivers provided responses.

**Table 9 nutrients-12-03136-t009:** Timeframe of GI problems in FXS study population as a function of soy-based diet.

Age	Soy % (*N* = 18)	No Soy % (*N* = 31)	*p*	Odds Ratio	95% CI
0–2 weeks (%)	61	9.7	0.0002 ^1^	15	3.2–67
0–12 months (%)	78	58	0.22 ^1^	2.5	0.67–9.5
0–3 years (%)	100	77	0.038 ^1^	11	0.61–211
Mean age (mo) (SEM)	7.8 (2.7)	52 (15)	0.031 ^2^	n/a	n/a

^1^ Fisher exact test. ^2^ Student *t*-test.

**Table 10 nutrients-12-03136-t010:** Timeframe of allergies in FXS study population as a function of soy-based diet.

Age	Soy % (*N* = 22)	No Soy % (*N* = 40)	*p*	Odds Ratio	95% CI
0–2 weeks (%)	14	0	0.041 ^1^	15	0.72–296
0–12 months (%)	45	20	0.035 ^2^	3.3	1.1–10
0–3 years (%)	77	60	0.17 ^2^	2.3	0.70–7.4
Mean age (mo) (SEM)	28 (6.7)	50 (7.2)	0.06 ^3^	n/a	n/a

^1^ Fisher exact test. ^2^ Chi square test. ^3^ Student *t*-test.

**Table 11 nutrients-12-03136-t011:** Allergens reported in FXS study population.

*N*	Allergen ^1^
44	Seasonal (dust, grass, hay, mold, pollen, trees, weeds)
31	Food products (corn, dairy, eggs, fish, food dyes/fillers, gluten, oats, oranges, peanut, rice, sesame, tree nuts, turkey, wheat)
20	Medications (Abilify, Amoxicillin, Brevital IV, cephalosporins, vaccines (Prevar, H1N1 flu), Risperdal, Ritalin, sulfa drug, Trileptal)
11	Animals (cats, dogs, feathers, hair, mice, wood clothing)
2	Insects
2	Chemicals (detergents, perfumes)

^1^ A total of 74 caregivers provided responses with many listing multiple allergens.

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
