# Peer review of "Soy-Based Infant Formula is Associated with an Increased Prevalence of Comorbidities in Fragile X Syndrome"

_nutrients, 2020, doi:10.3390/nu12103136_

Round 1
Reviewer 1 Report
The authors conducted a retrospective survey analysis to determine if there were associations between consumption of soy-based infant formula and seizure history, cognitive ability and autistic behaviors in participants with FXS. It would not be surprising to find an association, given many children with GI issues would be switched to soy-based products, and the fact that autism is associated with a higher incidence of GI issues and seizure disorders. This does not show causality, or cause and effect.
Thus, the study results only indicate the presence of an association, and this needs to be emphasized adequately. In the present version of the article, it is not adequately emphasized, and there seem to be statements suggesting cause and effect (eg, lines 93, 113), which is erroneous.
While the authors do address this in the discussion (eg, lines 426-429), it needs to be highlighted clearly that the study merely shows an association between soy-based infant formula and autism, etc in FXS- which in fact could very well be due to GI issues in such infants (with autism) being a confounding factor. This is especially true in this study because it is noted that the highest-ranked reason for the use of soy-based infant formula was GI / medical issues.
Author Response
We thank the reviewers for their careful critique of our work. Please find our responses below.
Reviewer 1
The authors conducted a retrospective survey analysis to determine if there were associations between consumption of soy-based infant formula and seizure history, cognitive ability and autistic behaviors in participants with FXS. It would not be surprising to find an association, given many children with GI issues would be switched to soy-based products, and the fact that autism is associated with a higher incidence of GI issues and seizure disorders. This does not show causality, or cause and effect.
Thus, the study results only indicate the presence of an association, and this needs to be emphasized adequately. In the present version of the article, it is not adequately emphasized, and there seem to be statements suggesting cause and effect (eg, lines 93, 113), which is erroneous. Thank you. We have changed the statement in line 95 from “effects on” to “associations between”. We have changed the statement in line 116 from “the effects of” to “associations between”.
While the authors do address this in the discussion (eg, lines 426-429), it needs to be highlighted clearly that the study merely shows an association between soy-based infant formula and autism, etc in FXS- which in fact could very well be due to GI issues in such infants (with autism) being a confounding factor. This is especially true in this study because it is noted that the highest-ranked reason for the use of soy-based infant formula was GI / medical issues. We have added the following statement, “We emphasize that this study shows associations between soy-based infant formula and FXS comorbidities and not cause and effect relationships”, to the end of the Introduction (lines 99-100) to clearly highlight that the study shows associations and not cause and effect. GI problems as a confounding issue are discussed lines 467-471.
Reviewer 2 Report
The studies of Westmark CJ et al. investigated associations between the consumption of soy-based formula during infancy and disease comorbidities in persons with fragile X syndrome (FXS) through a retrospective case-control survey study of participants with FXS enrolled in the Fragile X Online Registry with Accessible Research Database (FORWARD). Significant associations between the consumption of soy-based infant formula and the comorbidity of autism, 40 gastrointestinal problems (GI) and allergies were observed. The authors concluded that early-life feeding with soy-based infant formula is associated with the development of several disease comorbidities in FXS.
The sample size of the study is well justified and data collection in the context of rare disease was extremely well designed and performed. Since the studies of FXS mouse model (one study) suggest that soy-based diet exacerbates seizures, one of the main hypothesis of the studies was to study effects of the soy-based formula in human. Data from the survey regarding seizure type was not considered accurate and effects of soy-based formula on epilepsy in FXS was not confirmed. However, use of soy-based infant formula was shown to associate with gastrointestinal problems and autistic behavior. In these associations, there should be more evidence to support the causal effects of soy-based formula. It would have been surprising if the correlation between soy-based formula and gi-problems was not identified. The conclusion of the authors that an association with gi-problems and autistic features exist is correct but the title “with increased autism” is not relevant or should be supported with additional studies. In discussion the authors state that they “corroborate prior retrospective analysis of SFARI-SSC data” but this was not the case in reality.
The studies aimed to investigate neurodevelopmental effects of soy particularly due to its potential estrogen activity. This makes it necessary to analyze the effects in FXS females and males separately, too.
Author Response
We thank the reviewers for their careful critique of our work. Please find our responses below.
Reviewer 2
The studies of Westmark CJ et al. investigated associations between the consumption of soy-based formula during infancy and disease comorbidities in persons with fragile X syndrome (FXS) through a retrospective case-control survey study of participants with FXS enrolled in the Fragile X Online Registry with Accessible Research Database (FORWARD). Significant associations between the consumption of soy-based infant formula and the comorbidity of autism, 40 gastrointestinal problems (GI) and allergies were observed. The authors concluded that early-life feeding with soy-based infant formula is associated with the development of several disease comorbidities in FXS.
The sample size of the study is well justified and data collection in the context of rare disease was extremely well designed and performed. Since the studies of FXS mouse model (one study) suggest that soy-based diet exacerbates seizures, one of the main hypothesis of the studies was to study effects of the soy-based formula in human. Data from the survey regarding seizure type was not considered accurate and effects of soy-based formula on epilepsy in FXS was not confirmed. However, use of soy-based infant formula was shown to associate with gastrointestinal problems and autistic behavior. In these associations, there should be more evidence to support the causal effects of soy-based formula. It would have been surprising if the correlation between soy-based formula and gi-problems was not identified. The conclusion of the authors that an association with gi-problems and autistic features exist is correct but the title “with increased autism” is not relevant or should be supported with additional studies. We have changed the title to “Soy-Based Infant Formula is Associated with an Increased Prevalence of Comorbidities in Fragile X Syndrome.” In discussion the authors state that they “corroborate prior retrospective analysis of SFARI-SSC data” but this was not the case in reality. We modified this statement to clearly indicate that the SFARI seizure data was not statistically corroborated by the FORWARD study, but both studies did show associations between soy-based infant formula and allergies and autism. Lines 430-449.
The studies aimed to investigate neurodevelopmental effects of soy particularly due to its potential estrogen activity. This makes it necessary to analyze the effects in FXS females and males separately, too. We agree that the separate study of females and males as related to potential estrogenic activity would be interesting. However, there were only 10 females in the soy-based infant formula group making the study size to too small to draw meaningful conclusions based on sex.
Round 2
Reviewer 2 Report
The study of Westmark et al is comprehensive and well done. The final results showing some correlations between nutrients and certain symptoms in fragile X syndrome, however, could be due to increased clinical problems and need for tools to prevent them- not symptoms caused by nutrients. I feel that argument for increased symptoms caused by nutrients is misleading. In addition, although female subgroup is small. it would be important and interesting to analyze and show the data separately.
Author Response
The study of Westmark et al is comprehensive and well done. The final results showing some correlations between nutrients and certain symptoms in fragile X syndrome, however, could be due to increased clinical problems and need for tools to prevent them- not symptoms caused by nutrients. I feel that argument for increased symptoms caused by nutrients is misleading. In addition, although female subgroup is small. it would be important and interesting to analyze and show the data separately.
We thank the reviewers for the careful critique of our work. We were careful to avoid a “cause and effect” argument as we can only ascertain associations with a case-control survey study. In re-reading the paper, we adjusted vocabulary at lines 109, 408 and 410 to avoid any implication of cause, and with the last revision we had added the following to the Introduction, “We emphasize that this study shows associations between soy-based infant formula and FXS comorbidities and not cause and effect relationships.
We agree that it is important and interesting to analyze the data as a function of sex. We have added a Supplementary Table 1 with the data binned by males and females (line 303). The sample size is small (n=10) for female cohort on soy.